# Microbial Community in the Permafrost Thaw Gradient in the South of the Vitim Plateau (Buryatia, Russia)

**DOI:** 10.3390/microorganisms10112202

**Published:** 2022-11-07

**Authors:** Svetlana Zaitseva, Nimazhap Badmaev, Lyudmila Kozyreva, Vyacheslav Dambaev, Darima Barkhutova

**Affiliations:** Institute of General and Experimental Biology SD RAS, 670047 Ulan-Ude, Russia

**Keywords:** permafrost, microbial diversity, high-throughput Illumina sequencing, microbial ecology

## Abstract

Soil microbial communities play key roles in biogeochemical cycles and greenhouse gas formation during the decomposition of the released organic matter in the thawing permafrost. The aim of our research was to assess the taxonomic prokaryotic diversity in soil-ecological niches of the Darkhituy-Khaimisan transect during the initial period of soil thawing. We investigated changes in the microbial communities present in the active layer of four sites representing distinct habitats (larch forest, birch forest, meadow steppe and thermokarst lake). We explore the relationship between the biogeochemical differences among habitats and the active layer microbial community via a spatial (across habitats, and with depth through the active layer) community survey using high-throughput Illumina sequencing. Microbial communities showed significant differences between active and frozen layers and across ecosystem types, including a high relative abundance of *Alphaproteobacteria*, *Firmicutes*, *Crenarchaeota*, *Bacteroidota* and *Gemmatimonadota* in the active layer and a high relative abundance of *Actinobacteriota* and *Desulfobacterota* in the frozen layer. Soil pH, temperature and moisture were the most significant parameters underlying the variations in the microbial community composition. CCA suggested that the differing environmental conditions between the four soil habitats had strong influences on microbial distribution and diversity and further explained the variability of soil microbial community structures.

## 1. Introduction

Permafrost is distributed over an area of more than 22 million km^2^, occupying about 24% of the land in the northern hemisphere [1]. On the territory of Russia, 65% of landscapes are located in the area of permafrost [2]. Cryogenic territories occupy a small part in the soil cover of Eastern Siberia. In the south of the Vitim Plateau, they are found under the middle and northern taiga shrub–lichen–moss larch and birch–forb forests on rubble–loam deposits in a cold, humid, semi-humid and semi-arid climate [3].

The southern border of the permafrost zone passes through the territory of the south of the Vitim Plateau and the north of the Selenginsky midlands of Transbaikalia. This area has a high degree of diversity of soil and climatic conditions. For a century, the boundaries of permafrost receded. The depth of thawing of permafrost soils began to be found at about 3 m, against 2.5 m in the 70s of the last century, which was the result of serious climatic changes [4,5]. At the same time, the highest rate of permafrost retreat was observed in open steppe landscapes, while the lowest rate was observed in forest and meadow-marsh landscapes [6].

Permafrost is a very vulnerable biome, in which significant carbon reserves are buried in perennial and seasonally frozen soils [2]. Permafrost degradation makes this carbon available for utilization by the microbial community. The increased interest in the study of permafrost throughout the world, in particular, is due to the need to assess the greenhouse gases (CO_2_, CH_4_, N_2_0) formation during the microbial decomposition of the released organic matter. An inverse relationship has been established between climate warming and the gradual and long-term release of greenhouse gases from the soils of the Arctic and subarctic regions [7,8]. Recent studies of greenhouse gas emissions (CO_2_, CH_4_, N_2_0) from the soils of the Arctic zone of Russia and the ultracontinental regions of Siberia with permafrost have shown that the Siberian ultracontinental permafrost landscapes are characterized by aridity with sufficient heat supply and support higher soil CO_2_ emission rates, in spite of dryness, owing to the larger phytomass storage, presence of tree canopies, thicker active layer, and greater expressed soil fissuring [9]. The seasonal dynamics of soil CO_2_ production in meadow–chernozem permafrost soils in the south of the Vitim Plateau (Eravninskaya depression) and meadow long-seasonally freezing soils of the Selenga river delta mainly depend on soil temperature and only at the beginning of the growing season was a humidity influence observed [10].

The study of the taxonomic diversity of prokaryotes in permafrost soils is a necessary step to assess the role of microbial communities in specific environmental conditions, to suggest possible biogeochemical processes occurring in specific ecotopes in modern ecosystems under global warming and anthropogenic impacts and to predict the ecosystem responses to thaw.

Previous studies of the microbial component of the Transbaikalia soils by classical microbiological methods have established the number of metabolically important groups of microorganisms during the different seasons and their enzymatic activity in various types of soils [11,12,13]. However, the conducted studies do not characterize the taxonomic composition of microbial communities.

The Eravna Basin, bounded by the Vitim Plateau in the northeast and the Selenga Middle Mountains in the south, is a model system for microbial ecology studies in various soil-ecological niches. The object of our studies was the soil cover of the Darkhituy–Khaimisan transect (Eravninskaya depression, Buryatia). The soils of the transect polygon were well-characterized earlier [3,14].

The aim of the work was to assess the taxonomic prokaryotic diversity in various soil-ecological niches of the Darkhituy–Khaimisan transect during the initial period of soil thawing. We investigated changes in the active layer microbial community of four sites representative of distinct habitats (larch forest, birch forest, meadow steppe and thermokarst lake herb–sedge community) differing with permafrost thaw stages. Here we explore the relationship between the biogeochemical differences among habitats and the active layer microbial community via a spatial (across habitats, and with depth through the active layer) community survey using 16S rRNA gene amplicon sequencing. This is the first attempt to understand this impact by addressing the microbial diversity and distribution with respect to climate-induced thaw and correlated environmental parameters.

## 2. Materials and Methods

### 2.1. Soil Sampling and Analysis

We collected the seasonally thawed active layer of permafrost soil from the Darkhituy slope (1015 m a.s.l.) and an adjacent thermokarst dried-up lake Khaimisan from the southwestern part of the Vitim Plateau in the Yeravna Lowland (52°38′ N, 111°24′ E, Figure 1). The catena starts at the top of a mountain with larch taiga, crosses a gentle northeastern slope with a birch forest, and ends at the foothill plain (948 m a.s.l.). Soil samples were taken in mid-May 2021, during the initial period of thawing. At each site, three randomly chosen plots were selected as replicates. In each plot, possible depth cores were taken, and soil samples were collected from depth intervals representing the active layer, which is the surface-thawed soil at the time of sampling, and the permafrost layer, which is soil at or below the permafrost interface. The three subsamples from each depth were pooled into a single soil sample. Sampling depth and number of samples changed across habitats, resulting in 21 soil samples (Appendix A). Samples for molecular genetic studies were selected on the basis of previous morphological and geochemical studies [3,5,6,14]. Soil moisture, temperature and pH were synchronously measured using a mobile measurement system for the coupled monitoring of atmospheric and soil parameters (ASMC, Tomsk, Russia), as well as a pH meter (HM Digital, Seoul, Korea). The soil temperature at a depth of up to 75 cm was measured with a TK-5.05 electronic thermometer (0.1 °C resolution, ±1 °C accuracy). Soil ammonia nitrogen was determined by colorimetric methods with Nessler’s reagent, soil organic carbon was determined by Tyurin’s microchromic method [15]. The soil samples were transported to the laboratory while stored on ice as soon as possible. Table 1 provides a summary of sampling locations.

### 2.2. DNA Extraction, Amplification and Sequencing

To isolate DNA from soil samples, we used a reagent kit (MACHEREY-NAGEL NucleoSpin Soil) from MACHEREY-NAGEL (Düren, Germany) according to the manufacturer’s instructions.

Purified DNA preparations were used to create libraries of 16S rRNA gene fragments by PCR using universal primers for the V4 variable region: F515/R806 (GTGCCAGCMGCCGCGGTAA/GGACTACVSGGGTATCTAAT) [16], with attached adapters and unique Illumina barcodes. PCR was carried out in 15 μL of the reaction mixture containing 0.5–1 unit of activity of Q5 High-Fidelity DNA Polymerase (NEB, Ipswich, MA, USA), 5 pM each of forward and reverse primers, 1–10 ng of DNA template, and 2 nM of each dNTP (life technologies, Carlsbad, CA, USA). The mixture was denatured at 94 °C for 1 min, followed by 25 cycles: 94 °C for 30 s, 55 °C for 30 s, and 72 °C for 60 s. This final elongation was carried out at 72 °C for 3 min. PCR products were purified according to the method recommended by Illumina using AMPureXP magnetic particles (BeckmanCoulter, Brea, CA, USA). Library preparation and sequencing were carried out in accordance with the manufacturer’s recommendations for operation of the Illumina MiSeq instrument (Illumina, San Diego, CA, USA) using the MiSeq^®^ ReagentKit v3 (600 cycle) in a paired-end run (2 × 300 n). Initial data processing, namely, sample demultiplexing and removal of adapters, was carried out using Illumina software (Illumina, USA). The research was done using equipment of the Core Centrum ‘GenomicTechnologies, Proteomics and Cell Biology’ in ARRIAM (Saint-Petersburg, Russia). For subsequent denoising, sequence merging, deletion of chimeric reads, restoration of the original phylotypes (ASV, Amplicon sequence variant), and further taxonomic classification of the obtained ASVs, the software packages DADA2 [17], PHYLOSEQ [18] and SILVA [19] were used; the work was carried out in the R software environment. For the presentation of taxonomic analysis data, the tools of the QIIME 1 software package [20] were used.

### 2.3. Statistical Analysis

Statistical and mathematical processing of multidimensional data of environmental parameters of the studied lakes was carried out to graphically display the results and determine the parameters that are most typical for the ecosystem of each soil. Alpha diversity metrics including the Shannon index and Simpson’s diversity index were calculated using the MatLab11 package (The MatWorks, Inc., Beltsville, MD, USA). Pearson correlations were used to assess the associations between the microbial alpha diversity and environmental factors. A value of *p* less than 0.05 was considered to be statistically significant. Preliminary processing for data standardization was carried out according to the recommendations [21]. To control covarying effects of various factors, partial Mantel tests were performed using the MatLab11 package (The MatWorks, Inc.). Regression tree analysis was performed to identify differences in soil properties between the layers and between ecosystem types. The relationships between the relative abundance of the major microbial phyla, as well as the Shannon index and Simpson diversity index, with soil physicochemical parameters, were tested by Spearman’s correlation analysis. The relative importance of each individual environmental variable on bacterial community composition was measured by Principal Component Analysis (PCA) performed using the MatLab11 package (The MatWorks, Inc., Beltsville, MD, USA). A constrained ordination was carried out by a Canonical Correspondence Analysis (CCA) to correlate environmental variables with microbial taxonomic diversity and samples [22]. CCA was performed with XLSTAT (Addinsoft, Paris, France).

## 3. Results

### 3.1. Soil Environmental Parameters

At the time of sampling, the least soil thawing was observed in site D1, where seasonal permafrost was recorded at a depth of 17 cm. At Khaimisan sampling location frozen soil observed from a depth of 67 cm. In general, soil properties changed across habitats (Table 1 and Table 2, Figure 2 and Appendix A). The soil was slightly acidic and neutral on the Darhituy slope and slightly alkaline in the Khaimisan sample, but the pH values did not change significantly with depth. Soil temperature sharply decreased with depth, from 9.6 °C in the active thawing layer to −0.5 °C in the permafrost layer. Soil moisture slightly increased with depth on the Darhituy catena; significant moisture variations were noted along the horizons depth in the Khaimisan sample. The highest humidity (110%) was preserved under the forest cover. Soil total organic carbon (TOC) varied significantly across habitats and decreased significantly with depth (*p* < 0.05). Ammonium nitrogen (N/NH_4_) content was highest in the subsurface layer at the Khaimisan sampling location. Nitrate nitrogen (N/NO_3_) content decreased with depth; the maximum content was found in the birch forest soil surface layer at site D2.

Regression tree analysis demonstrated the relative importance effects of individual environmental factors influencing the habitats’ similarity and clustering of sampling sites (Appendix A). The most important environmental parameter in sample clustering was the soil horizon depth, which was mainly determined by the depth of the thawing horizon and active layer thickness.

### 3.2. Soil Microbial Community Structure and Diversity

In total, more than 380 thousand reads of 16S rRNA gene fragments were obtained; 58% (220,784) of the reads were classified to at least the kingdom level. In further analysis, only classified sequences were used; their share of the total number of reads obtained was 48.9–71.3% (Table 3). The amplicon sequence variants, ASVs, varied from 214 to 533. Relatively more ASVs were observed in the surface layer under the forb-sedge community at Khaimisan sampling location (Kh-1), as well as under the birch forest (D2-1, D2-2). In all samples, a decrease in the number of ASVs towards the lower horizons was noted. The smallest amount was recorded in the permafrost under the larch forest (D1-2). Diversity assessment using diversity indices showed that both species richness (number of ASVs) and Shannon and Simpson diversity indices decreased in soils under larch and birch forests in frozen soil. In the meadow steppe core, the species richness was higher in the subsurface layer, and the diversity indices increased.

**Table 3 microorganisms-10-02202-t003:** Soil microbial community diversity.

Sample	Number of Classified Reads/Share, %	ASVs	Shannon Diversity Indices	Simpson Diversity Indices	Simpson Invers Diversity Indices
Site D1-1	15,023/48.94	248	4.62	0.975	40.599
Site D1-2	16,288/53.52	214	4.35	0.974	38.11
Site D2-1	23,363/71.29	431	5.28	0.988	86.33
Site D2-2	18,774/58.45	391	5.08	0.982	56.457
Site D3-1	19,000/60.37	329	4.73	0.972	35.17
Site D3-2	15,939/53.1	333	4.77	0.973	37.445
Site D3-3	15,829/52.08	323	4.85	0.980	49.459
Site Kh-1	20,046/59.92	533	5.62	0.990	104.01
Site Kh-2	19,085/59.4	370	4.89	0.976	42.515
Site Kh-3	18,920/60.27	315	4.58	0.973	37.541
Site Kh-4	19,574/54.71	316	4.22	0.943	17.667
Site Kh-5	18,943/62.37	274	3.67	0.892	9.259

with depth. In the Khaimisan lake core, with depth, there is a sharp decrease in both species diversity and the values of diversity indices. The Shannon and Simpson diversity indices varied from 3.67 to 5.62 and from 9.3 to 104, respectively. The Shannon and Simpson diversity indices showed significant differences, suggesting that diversity of the microbial 16S rRNA gene libraries differs among the layers and across habitats (Table 3, Figure 3).

**Figure 3 microorganisms-10-02202-f003:**
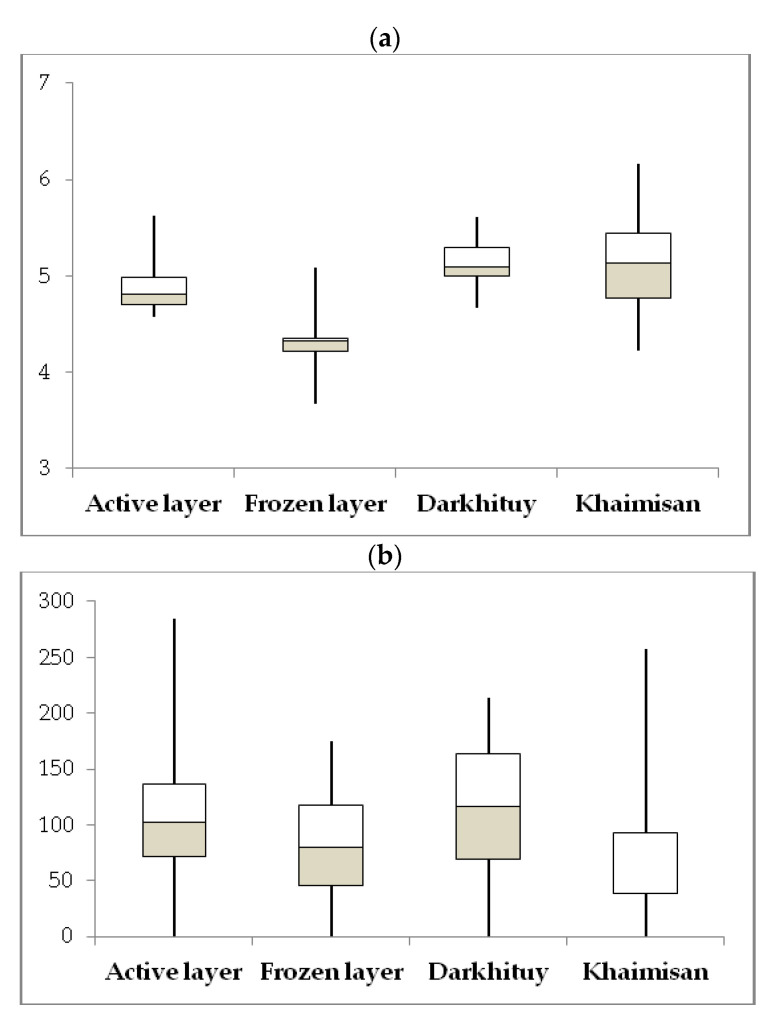
Boxplots of differences in alpha diversity between the active and frozen layers and across habitats in Darkhituy–Khaimisan. All data are presented as the mean ± standard error. (**a**) Shannon diversity index; (**b**) Number of ASVs.

The most dominant microbial phyla and proteobacterial classes (more than 1% of total sequences) detected across all soil samples were *Actinobacteriota*, accounting for ~28% of all sequences, followed by *Alphaproteobacteria*, *Gammaproteobacteria*, *Firmicutes*, *Crenarchaeota*, *Acidobacteriota*, *Bacteroidetes*, *Verrucomicrobia*, *Halobacterota* and *Planctomycetes* (Appendix A, Figure 4). The relative abundance of these dominant phyla showed different layer-associated patterns across all habitats. Furthermore, the relative abundance of the dominant bacterial phyla varied significantly (*p* < 0.05) between sample sites (Appendix A), with *Firmicutes*, *Alphaproteobacteria*, *Actinobacteriota*, *Acidobacteriota* and *Verrucomicrobiota* being more abundant in the soil sample of the larch forest; *Crenarchaeota* was the most abundant in the samples of the steppe meadows and the larch-birch forest, whereas thermokarst lake Khaimisan soil samples were dominated by *Actinobacteriota*, *Alphaproteobacteria*, *Gammaproteobacteria*, *Halobacterota*, *Bacteroidota*, *Firmicutes*, *Acidobacteriota* and *Verrucomicrobiota*.

The archaeal community was the most diverse and abundant in the thermokarst lake Khaimisan soil samples. The active layer in the bottom of dried-up lake Khaimisan contained a proportion of *Crenarchaeota* from 0.1% to 2.4%, compared to up to 1.7% *Euryarchaeota* and 3.8–16.5% *Halobacterota*, whereas the permafrost had a high abundance of *Halobacterota* only (1–3%). Other archaeal phyla were not found, or their share was less than 1%. The composition of archaeal communities of Darkhituy slope demonstrated that they were only affiliated with the phylum *Crenarchaeota* and belonged mainly to unclassified ammonia-oxidizing archaea (AOA), *Nitrososphaeraceae,* or rarely two Candidatus genera: “*Ca*. Nitrocosmicus” and “*Ca*. Nitrososphaera”.

Microbial communities showed significant differences at the phylum level between active and frozen layers, including a high relative abundance of *Alphaproteobacteria*, *Firmicutes*, *Crenarchaeota*, *Bacteroidota* and *Gemmatimonadota* in the active layer (*p* < 0.05) and a high relative abundance of *Actinobacteriota* and *Desulfobacterota* in the frozen layer (*p* < 0.05) (Figure 5a). The relative abundances of *Firmicutes*, *Verrucomicrobiota*, *Gemmatimonadota*, *Planctomycetota* and noticeably *Crenarchaeota* in the soil microbial communities of Darkhituy were significantly higher than those in Khaimisan (*p* < 0.05), whereas the relative abundance of *Actinobacteriota*, *Gammaproteobacteria* and *Halobacterota* showed the opposite trend (Figure 5b). The influence of distinct habitats (larch forest, birch forest, meadow steppe and thermokarst lake herb-sedge community) was apparent at the class, family and genus level (Appendix A). For instance, the relative abundances of phenotypically, metabolically, and ecologically diverse *Xanthobacteraceae*, *Clostridiaceae* and globally distributed *Chthoniobacteraceae* were more than 5% of total microbial diversity in both active and frozen layers of D1 soil sample under larch forest. At the same time, 12 and 17 ASVs associated with *Chthoniobacteraceae* were detected in the active and frozen layer, respectively, ten of which were found in the active and frozen layers in different proportions; the others were present only in one of the layers.

One of the most abundant genera was *Bradyrhizobium*, soil nitrogen-fixing bacteria free-living or associated with the roots of leguminous plants. Although *Xanthobacteraceae*, *Chthoniobacteraceae* and *Clostridiaceae* were rather abundant at the D2 site (birch forest), ammonia-oxidizing archaea *Nitrososphaeraceae* dominated in the frozen layer and amounted to more than 22% of total abundance. The active layer at D2 site was characterized by a wide variety of taxa on the genus level, such as ASVs affiliated with *Mycobacterium* (1.65%) (*Corynebacteriales*; *Mycobacteriaceae*), *Kribbella* (1.04%) and *Nocardioides* (1.36%) (*Propionibacteriales*; *Nocardioidaceae*). Ammonia-oxidizing archaea *Nitrososphaeraceae* amounted 14–29% at the D3 site (meadow steppe) and decreased with soil depth, whereas the total number of ASVs affiliated with the phylum *Actinobacteriota* increased with depth and notably varied in the composition of dominant genera. *Nocardioidaceae* (10%), mainly presented by g. *Nocardioides*, and *Propionibacteriaceae* (4.86%) of g. *Microlunatus* dominated in the surface layer. The most abundant *Actinobacteriota* in deeper layers on the genus level were unclassified *Micrococcaceae* (more than 8%) and *Microbacteriaceae* (4.7%; 7.8%), mainly affiliated with g. *Agromyces* (Appendix A). *Verrucomicrobiota* (7.77%; 7.05%; 4.7%), in sites D1 and D2 samples, were mainly affiliated with “*Candidatus* Udaeobacter” (*Verrucomicrobiae*; *Chthoniobacterales*; *Chthoniobacteraceae*). *Planctomycetota* amounted from 1 to 4% in all Darkhituy sites and decreased with soil depth at sites D1 and D2, while in the meadow steppe (D3), on the contrary, it slightly increased. There was a difference in the dominant families. Thus, in site D1, ASVs affiliated with *Isosphaeraceae* dominated, *Isosphaeraceae* and *Pirellulaceae* were the most abundant in sites D2, and only *Pirellulaceae* was numerous in site D3. The number of dominant microbial taxa (accounted more than 1% of total diversity) declined dramatically with depth, from 27 in the surface layer to 11 in the deep frozen layer at the thermokarst lake Khaimisan soil. However, *Actinobacteriota* presented by *Micrococcaceae*, *Microbacteriaceae* and *Nocardioidaceae* increased sharply in deeper layers. Anaerobic methanotrophic archaea (ANME) belonging to the archaeal family *Methanoperedenaceae* and “*Candidatus* Methanoperedens” accounted for more than 11% of total diversity in subsurface layer and were also numerous in the deep frozen layers.

### 3.3. Relationships between Environmental Variables and Microbial Community Structures

Whole microbial community analysis by canonical correspondence analysis (Figure 6) clustered samples by sites. Samples separated along the primary CCA axis according to pH states with more alkaline Khaimisan samples clustered together right of the origin and the neutral Darkhituy samples to the left. The secondary axis separated samples according to depth from surface site samples diverging from each other with depth. The first two axes accounted for 44% of the variance for microbial community.

Generally, an increase in soil pH resulted in a higher diversity and number of the archaea *Halobacterota*, *Euryarchaeota*, *Thermoplasmatota,* as well as of the bacteria *Chloroflexi*, *Fusobacteriota*, *Gammaproteobacteria* and *Nitrospirota* (Figure 6). CCA showed that the abundance of *Euryarchaeota, Thermoplasmatota*, *Fusobacteriota*, *Bacteroidota* and *Myxococcota* was positively influenced by higher soil temperatures. A higher depth exhibited an increase in the relative abundance of microbial phyla *Actinobacteriota* and *Fibrobacterota*. In contrast, *Crenarchaeota* was positively influenced by the higher total organic matter content.

We also measured the relative importance of each single environmental variable for microbial community composition. Of all the soil environmental factors examined, soil pH, soil temperature and moisture were the most significant parameters underlying the variations in the microbial community composition. The combination of these variables explained ~30% of the observed variation in soil microbial community composition. The relative contributions of depth and permafrost explained ~14% of the observed variation in microbial diversity (Table 4). The effects of soil physicochemical factors on bacterial diversity on families level were tested by Spearman’s correlation analysis (Appendix A). The Shannon index was significantly positively correlated with soil temperature only, and was negatively correlated with soil depth (*p* < 0.05). The Simpson diversity index was significantly negatively correlated with soil depth. Representatives of *Alphaproteobacteria* from order *Rhizobiales*, family *Xanthomonadaceae* (r^2^ = 0,6), *Hyphomicrobiaceae* (r^2^ = 0.6), *Labraceae* (r^2^ = 0.4), *Planococcaceae* (*Firmicutes*; *Bacilli*; *Bacillales*) (r^2^ = 0.44), *Spirosomaceae* (*Bacteroidota*; *Bacteroidia*; *Cytophagales*) (r^2^ = 0.43), *Crocinitomicaceae* (*Bacteroidota*; *Bacteroidia; Flavobacteriales*) (r^2^ = 0.43), *Rubrobacteriaceae* (*Actinobacteriota; Rubrobacteria; Rubrobacterales*) (r^2^ = 0,37) and archaea *Nitrososphaeraceae* (*Crenarchaeota*; *Nitrososphaeria*) (r^2^ = 0.34) were significantly positively correlated with TOC (*p* < 0.05). Soil ammonium nitrogen (N/NH_4_) level played important role for archaea *Methanoperedenaceae*, unclassified representatives of *Methanosarciniales* and bacteria *Weeksellaceae*, *Hydrogenophilaceae*, *Nitrosomonadaceae*, *Hymenobacteraceae*, *Desulfocapsaceae* and *Solirubrobacteraceae* distribution. Significant positive correlations between soil nitrate nitrogen (N/NO_3_) content and relative numerous bacterial families, *Mycobacteriaceae*, *Isosphaeraceae*, *Bacillaceae*, *Micromonosporaceae*, *Ilumatobacteracea,* were observed. However, the greatest influence N/NO_3_ had was on the rare bacterial taxa *Alcaligenaceae*, *Sporichthyaceae*, *Inquilinaceae*, *Streptosporangiaceae*, *Micropepsaceae* and *Kineosporiaceae*.

## 4. Discussion

The soils of the Eravna Basin in the south of the Vitim Plateau are characterized as soils of the transitional zone with sharp changes in the temperature regime, in contrast to the soils of more northern regions with a more constant temperature regime [5]. The active layer, which is a seasonally thawing surface and subsurface soil layer, suffers from repeated environmental disturbances due to frozen–thaw cycles [23,24]. During the summer period, soils within the investigated Darkhituy–Khaimisan transect are also showing a large temperature gradient along their depths profiles [5,6]. The active layer thickness differs across the sampling sites and was a significant factor determining bacterial and archaeal diversity. Our results also showed that the microbial community structures differ significantly between the active and frozen layers and across ecosystem types. This result was supported by several studies which reported that microbial communities differed significantly between the active layer and permafrost layer on the Tibetan Plateau [25,26] and in the Arctic region [23,27] and across different habitats [24,28,29]. Our results showed that soil microbial communities at the Eravna Bazin (Vitim Plateau) were dominated by *Actinobacteriota* and *Alphaproteobacteria*, which were partly inconsistent with several studies, which reported that bacterial communities were dominated by *Actinobacteriota* [30,31,32], *Proteobacteria* [29,33,34], *Bacteroidetes* [35] or *Proteobacteria*, *Acidobacteriota* and *Actinobacteriota* [36]. We suppose that these differences are due to the unique conditions of soil formation (mountainous relief, sharply continental climate, permafrost) and the properties of the studied soils, which are formed on the southern border of the permafrost zone of Transbaikalia [3,6,14].

Up to now the majority of studies on permafrost-affected soil microbial communities revealed the distribution patterns and drivers of soil microbial diversity in different types of permafrost regions, such as Alaska [27,35], the Canadian Arctic [23,30,31], Siberia [36,37,38], Greenland [39], Antarctica [40] and the Tibetan Plateau [28,29]. The abundance, richness, and evenness of communities generally decrease with soil depth [27,28,29,36,37,38,39,40], but there is still microbial activity at subzero temperatures and in the permafrost layer. In our investigation, environmental conditions, including depth, soil pH, soil temperature and humidity, nutrient contents of TOC, soil ammonium and nitrate nitrogen, all played certain roles in determining microbial community composition in the active and frozen layers of permafrost soils. Of all examined factors, soil pH and temperature were the main factors that influenced community composition and diversity patterns. The soil pH is known to have a strong effect on the structure of soil microbial communities at all scales [41,42,43,44] and also in the permafrost ecosystems of Arctic and Alpine regions [45,46,47,48] in permafrost-affected soils in northeastern China [28]. As the most imperative characteristic of soil, pH influences soil microorganisms by modifying enzymes’ activity as well as by controlling the accessibility of nutrient and moisture supplements through deciding the ionization balance in soil [28,49]. For example, pH decides the community composition and biogeography of ammonia oxidizers by influencing the ionization equilibrium of nitrate and ammonia in soils [50]. Our results indicated the relationship between soil pH and the relative abundance of dominant and rare bacterial groups such as *Nitrospirota*, *Gammaproteobacteria*, *Gemmatimonadota*, *Verrucomicrobiota* and *Chloroflexi*, as well as the archaea *Halobacterota*, mainly affiliated with ammonia-oxidizing *Methanoperedenaceae*. Previous studies demonstrated that *Acidobacteriota* favored acidic conditions [36] and were negatively correlated with pH, whereas *Proteobacteria* dominated in higher pH soils [41,45,49]. In our samples, *Acidobacteriota* were found in both slightly acidic and slightly alkaline soils, but it was noted that with increasing pH, the number of *Acidobacteriota* decreased mainly due to cl. *Acidobacteriae*, while the number of cl. *Blastocatellia* did not change significantly, remaining the dominant class of the phylum in all samples. These results fit well with previous studies in tundra soils [51].

Soil ammonia-oxidizing archaea are highly abundant and key players in global biogeochemical cycles [52]. Our results revealed a very high proportion of *Nitrososphaeraceae* (up to 29% in D3), higher than previously reported. Thus, in some soils, AOA can account for more than 10% of the total prokaryotic community [51,52]. Archaeal ammonia oxidizers should be expected to be more abundant in soils with high organic matter content, providing a constant source of substrate [53]. We assume that the soil of D3, which is available for grazing, constantly absorbs fresh organic matter with waste. While ammonia from mineralization of organic matter is released slowly and continuously, resulting in low but constant levels of ammonia, the application of mineral nitrogen fertilizers contributes to the release of ammonia. It can be assumed that there are different sources of ammonia at sites D2 and D3. Genomic and proteomic studies of *Nitrososphaera viennensis* notice that the ability to perform chemolithoautotrophic growth due to ammonia oxidation and fixation of CO_2_ and toxic compounds, differs from specific metabolic innovations associated with *Nitrososphaerales,* mediating growth and survival in the soil environment, including the ability to form biofilms, modify cell surface and cell adhesion, as well as carbohydrate conversion and the detoxification of aromatics and drugs [54]. Different studies have identified multiple soil factors that trigger the abundance of AOA [52]. These factors include pH, concentration of available ammonia, organic matter content, moisture content, nitrogen content, clay content, as well as other triggers.

Our results demonstrated that the environmental conditions of a particular habitat to a greater extent determined the microbial community composition. The most important environmental factors were pH and temperature, and, to a lesser extent, humidity and depth of the soil horizon. A significant proportion of taxa involved in nitrogen cycles (nitrogen-fixing, reducing nitrites and nitrates, oxidizing ammonium) and carbon, including those oxidizing methane under anaerobic conditions, has been established.

## 5. Conclusions

The first attempt to assess taxonomic diversity of microbial communities at permafrost soils in the south of the Vitim Plateau revealed significant differences in microbial community composition and diversity among soil sites at the Darkhituy–Khaymisan transect during the initial period of soil thawing. We suggested that the differing environmental conditions between the four soil habitats had strong influences on microbial distribution and diversity and further explained the variability of soil microbial community structures. The effort to profile the vertical distribution of microbial communities in the active and frozen layers enabled better evaluations of changes in microbial dynamics in response to permafrost thaw.

## Figures and Tables

**Figure 1 microorganisms-10-02202-f001:**
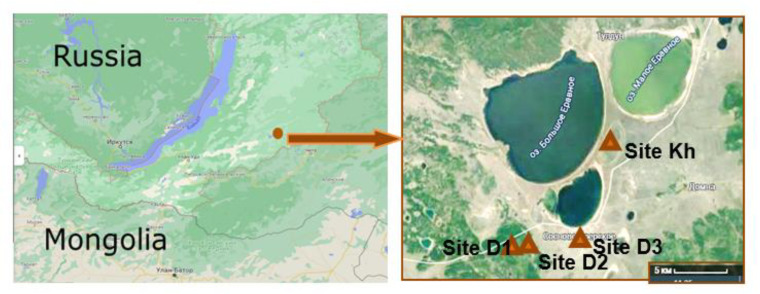
Map-scheme of the sampling sites along the Darkhituy–Khaimisan transect on the Vitim Plateau (Buryatia, Russia). At the right part of the map, the scale is 1:500,000.

**Figure 2 microorganisms-10-02202-f002:**
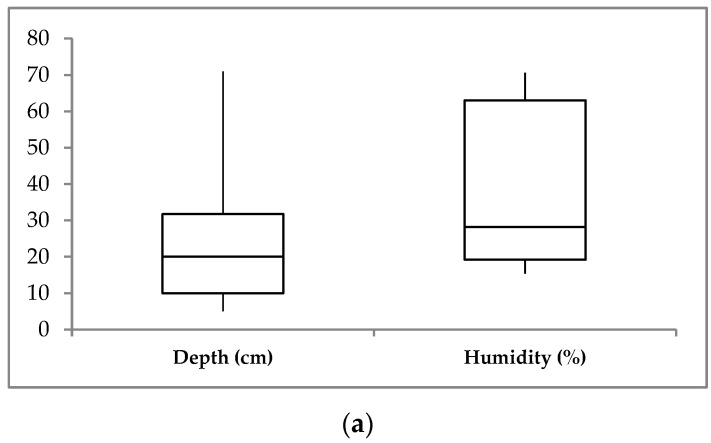
Boxplots of environmental variables for the sampling sites. (**a**) Depth (cm) and humidity (%) of soil sampling; (**b**) TOC (%), soil total carbon; N/NH_4_ (mg/100 g fresh soil), soil ammonium nitrogen; N/NO_3_ (mg/100 g fresh soil), soil nitrate nitrogen.

**Figure 4 microorganisms-10-02202-f004:**
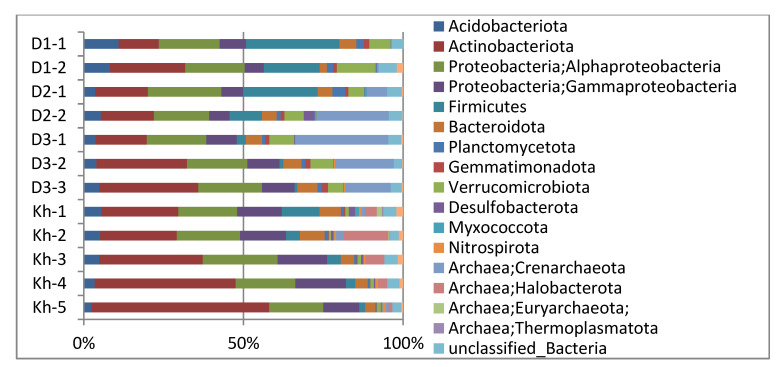
Relative abundance of bacteria and archaea on the phylum level in permafrost soils along the Darkhituy–Khaimisan transect.

**Figure 5 microorganisms-10-02202-f005:**
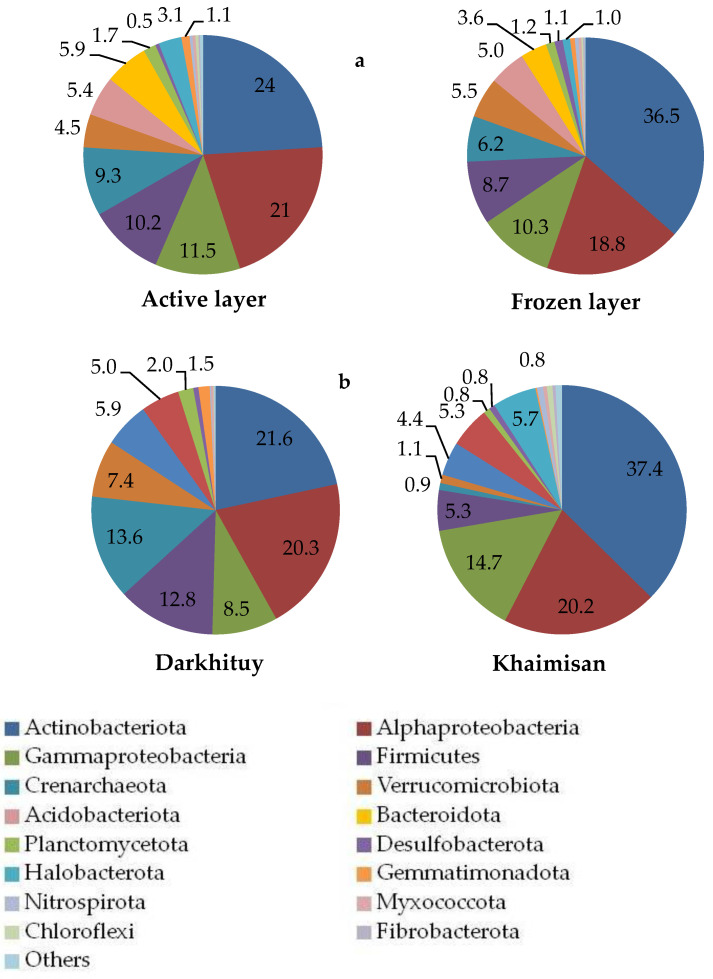
The relative abundance (%) of dominant microbial phyla and proteobacterial classes in the active and frozen layers (**a**); The relative abundance (%) of dominant microbial phyla in the Darkhituy and Khaimisan ecosystems (**b**). All data are presented as the mean. “Others” include phyla with a relative abundance < 1.0.

**Figure 6 microorganisms-10-02202-f006:**
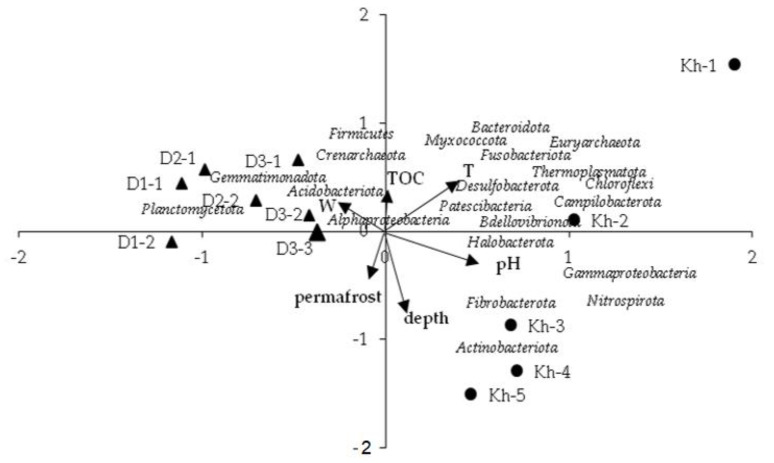
Canonical correspondence analysis (CCA) based on the microbial community structures and environmental factors (W−soil humidity, TOC−soil total carbon, T−soil temperature, permafrost—frozen layer: +1, or thawed layer: −1).

**Table 1 microorganisms-10-02202-t001:** Characteristics of sampling sites by [3,14].

	Darkhituy			Khaimisan
	Site D1	Site D2	Site D3	Site Kh
Coordinates *	52°31.222′ N, 111°26.346′ E	52°31.506′ N, 111°28.226′ E	52°030.905′ N, 111°032.770′ E	52°035.677′ N, 111°035.611′ E
Parent rock (according to genesis)	Eluvium; eluvio- deluvium	deluvium; deluvium carbonate	deluvium carbonate, deluvium, lacustrine deposits	lacustrine deposits
Vegetation	larch forest with *Rhododendron dauricum*, *Vaccinium vitisidaea, Pirola rotundifolia, Calamagrostis epigeios*	larch-birch forest with *Carex pediformis, Sanguisorba officinalis, Calamagrostis epigeios, Achillea* and parches of moss	meadow steppe with *Carex duriuscula, Sanguisorba officinalis, Artemisia mongolica*	herb-sedge community
a.s.l., m *	1052	965	957	944
Position in landscape (facies)	eluvial, transeluvial	transeluvial; transaccumula- tive	Transaccumula-tive; accumulative	accumulative
Thawing depth, cm (for the season)	270–300	230–250	275–290	170–200
WRB	Skeletic Cambisols (Gelic)	Cambic Calcic Cryosols (Siltic, Humic)	Eutric Chernozems Calcaric	Calcaric Humic Fluvisols Limnic Turbic

*—our data.

**Table 2 microorganisms-10-02202-t002:** Soil physicochemical characterization.

Site	Sample	Layer, cm	Color	T, °C	W, %	pH	TOC, %	N-NH_4_, mg/100 g	N-NO_3_, mg/100 g
D1	D1-1	5–15	brown	1.6 ÷ 0.1	65.86	5.97	1.59	28.0	6.0
	D1-2	17–25	dark brown	−0.5	62.56	6.1	0.81	22.3	1.8
D2	D2-1	5–15	almost black	2.0 ÷ 0.2	64.43	6.16	7.38	31.0	9.2
	D2-2	27–37	dark brown	−0.4	70.61	7.16	0.66	17.7	3.6
D3	D3-1	5–10	black	4.0	18.35	6.88	11.36	23.0	4.8
	D3-2	10–20	yellow brown	2.1	19.05	7.05	8.58	18.7	3.2
	D3-3	20–30	light brown	0.9	19.23	7.2	4.9	13.3	3.9
Kh	Kh-1	0–10	black	9.6 ÷ 7.5	41.97	8.91	6.2	16.7	3.7
	Kh-2	10–30	black brown	4.1 ÷ 2.1	23.6	8.9	2.3	58.3	5.3
	Kh-3	30–50	black brown	1.9 ÷ 1.3	15.36	8.62	2.47	n.d.	n.d.
	Kh-4	50–70	grey-yellow	0.2	32.72	8.62	2.3	25.7	2.8
	Kh-5	68–75	grey-yellow	−0.2	22.12	8.8	2.23	n.d.	n.d.

n.d.—not determined.

**Table 4 microorganisms-10-02202-t004:** Principal component analysis showing the principal components (PC), their explained variance and loadings of the parameters used in the PCA ordination.

Parameter	Abbreviated in Figure 6	PC1 (29.8%)	PC2 (13.9%)	PC3 (11.7%)
Depth	depth		−0.72	
Soil temperature	T	0.48		
Humidity	W	−0.41		0.61
pH	pH	0.57		
Total organic carbon	TOC			−0.54
Permafrost	permafrost		−0.43	

Variables not of importance for the explanation were omitted.

## Data Availability

The DNA sequences in this study have been deposited in the National Center for Biotechnology Information (NCBI) under BioProject PRJNA849711 and BioSample accession SAMN29127210.

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
