# Peer review of "Microbial Community in the Permafrost Thaw Gradient in the South of the Vitim Plateau (Buryatia, Russia)"

_microorganisms, 2022, doi:10.3390/microorganisms10112202_

Round 1
Reviewer 1 Report
Zaitseva et al. have submitted a manuscript entitled “Microbial community in permafrost thaw gradient in the south of the Vitim Plateau (Buryatia, Russia)”. The authors analyze bacterial communities as well as chemical properties of thawed active layer and permafrost soil of two distinct sites in the Vitim Plateau. The motivation and major research questions are clear, however there are several points that would benefit from clarification.
Major comments
Please make it clearer, in the Material and Methods section, how many samples were taken per site and which ones were representing active layer and permafrost. Please also explain what the “in-situ soil testing device” is or incorporate a reference for this. Please also mention which database was used for taxonomic affiliation of the ASVs.
In line 236-238 you mention that Chthoniobacteraceae were prevalent in both active and frozen layers of D1. Were these represented by the same ASVs?
In lines 288-294, you mention that you measure the relative importance of each variable on microbial community composition, but I did not understand how you did this and where I can see these results (for example, where can I see that these explain 30% of the variation?). Please clarify.
In lines 333-338 you mention that the bacterial phyla which dominated the samples were partly different from the ones reported from other studies. Do you have an explanation for this?
In line 362, you mention that Acidobacteriota are negatively correlated with pH, but in recent work and even in one of the references you listed (51) it is mentioned that distinct groups of Acidobacteriota have distinct pH preferences and this negative correlation with pH is mainly reported for members of the class Acidobacteriia. This nicely fits also to your observations so please adjust this section to reflect this.
Minor comments
Please check the naming of bacterial taxa throughout the manuscript: you should italicize all genera names and keep a consistency on the nomenclature (for example you have entries for both Acidobacteria and Acidobacteriota, which denote the same taxa).
Line 11: replace “…changes in the active layer microbial community of four sites representative distinct habitats…” with “…changes in the microbial communities present in the active layer of four sites representing distinct habitats…”
Line 27: replace “Permafrost is distributed over more than 22 million km2 area …” with “Permafrost is distributed over an area of more than 22 million km2”.
Line 43: replace “substance” with “carbon”.
Line 73: replace “are” with “were” .
Line 77: replace “representative distinct habitats” with “representative of distinct habitats” .
Line 118: replace “with double-sided reading” with “in a paired-end run” .
Figure 1: Please add a scale bar to the maps. Also, explain what the left and right panels are.
Line 159-160: You mention that a humidity of 110% was detected under the forest cover, but I cannot see these data anywhere in Table1, 2, Figure 2 and Figure S1.
Line 161: You mentioned that TOC varied significantly between habitats and depth, but I did not see any significance test. Please clarify.
Figure 2: Please indicate the units for all variables.
Line 177: replace “which mainly” with “which were mainly” .
Line 199: replace “The most dominant microbial phyla” with “The most dominant microbial phyla and proteobacterial classes” , wince later you mention Alpha- and Gammaproteobacteria.
Line 217: replace “from less of 0.1% to 2.4%” with “from 0.1% to 2.4%”.
Line 219: replace “Another” with “other”.
Line 221: replace “phyla” with “phylum”.
Line 249: replace “on the genus level, ASVs affiliated with…” with “on the genus level, such as ASVs affiliated with…”.
Line 254: Please explain what you mean with varied in dominant composition or rephrase this sentence.
Line 267: You mention dominant microbial taxa. At which level is this, genus?
Line 272: replace “numerouse” with “numerous”.
Figure 5: Please explain what the arrow labelled “permafrost” means in this context.
Line 302: please indicate the number of the supplementary table.
Line 317: replace “had on rare bacterial taxa” with “had was on rare bacterial taxa”.
Line 330: replace “significantly differ” with “differ significantly”.
Line 344: Please include a suitable reference.
Line 370: Please include a suitable reference.
Line 401: replace “significantly differ” with “differ significantly”.
Reviewer 2 Report
The manuscript is well written and it adresses important issue of microbial community dynamics in poorly studied permafrost soils.
My only major comment is that the authors should attempt more in-depth analysis of the link between micorbial diversity and physico-chemical properties of studied soils under similar context; see examples in:
Aksenov, A.S.; Shirokova, L.S.; Kisil, O.Y.; Kolesova, S.N.; Lim, A.G.; Kuzmina, D.; Pouillé, S.; Alexis, M.A.; Castrec-Rouelle, M.; Loiko, S.V.; et al. Bacterial Number and Genetic Diversity in a Permafrost Peatland (Western Siberia): Testing a Link with Organic Matter Quality and Elementary Composition of a Peat Soil Profile. Diversity 2021, 13, 328. https://doi.org/10.3390/d13070328
Another comment is that presumably, there is no duplicates. A study without duplicate soil samples can be easily rejected from any Q1 journal. I leave it at the editor discretion, whether it s applied in this particular case.
Specific comments
L98: -How much time it took for the transport?
Table 1: How many soil samples from each site; how many horizons? Replicates? Composite sample?
Fig 2 is not needed: it actually duplicates the content of the table
